

# Computational testing for automated preprocessing: a Matlab toolbox to enable large scale electroencephalography data processing

Benjamin U. Cowley[1,2], Jussi Korpela[1] and Jari Torniainen[3]

[1] BrainWork Research Centre, Finnish Institute of Occupational Health, Helsinki, Finland
[2] Cognitive Brain Research Unit, Faculty of Medicine, University of Helsinki, Helsinki, Finland
[3] Biophysics of Bone and Cartilage Group, Department of Applied Physics, University of Eastern Finland, Kuopio, Finland

## ABSTRACT

Electroencephalography (EEG) is a rich source of information regarding brain function. However, the preprocessing of EEG data can be quite complicated, due to several factors. For example, the distinction between true neural sources and noise is indeterminate; EEG data can also be very large. The various factors create a large number of subjective decisions with consequent risk of compound error. Existing tools present the experimenter with a large choice of analysis methods. Yet it remains a challenge for the researcher to integrate methods for batch-processing of the average large datasets, and compare methods to choose an optimal approach across the many possible parameter configurations. Additionally, many tools still require a high degree of manual decision making for, e.g. the classification of artefacts in channels, epochs or segments. This introduces extra subjectivity, is slow and is not reproducible. Batching and well-designed automation can help to regularise EEG preprocessing, and thus reduce human effort, subjectivity and consequent error. We present the computational testing for automated preprocessing (CTAP) toolbox, to facilitate: (i) batch-processing that is easy for experts and novices alike; (ii) testing and manual comparison of preprocessing methods. CTAP extends the existing data structure and functions from the well-known EEGLAB toolbox, based on Matlab and produces extensive quality control outputs. CTAP is available under MIT licence from https://github.com/bwrc/ctap.

## INTRODUCTION

Measurement of human electroencephalography (EEG) is a rich source of information regarding certain aspects of brain functioning, and is the most lightweight and affordable method of brain imaging. Although it can be possible to see certain large effects without preprocessing at all, in the general-case EEG analysis requires careful preprocessing, with some degree of trial-and-error. Such difficult EEG preprocessing needs to be supported with appropriate tools. The kinds of tools required for signal processing depends on the properties of data, and the general-case properties of EEG are demanding: large datasets and indeterminate data contribute to the number and complexity of operations.

Corresponding author
Benjamin U. Cowley,
benjamin.cowley@ttl.fi

In most research applications EEG data can be very large; systems are available with over 256 channels. This can result in the need to examine thousands or tens of thousands of data-points; for instance, visual examination of raw data quality for 50 subjects $\times$ 256 channels $\times$ 1,200 s $\cong$ 16,000 plot windows (where each window shows 32 channels $\times$ 30 s). Also, normally EEG can require many operations (see e.g. *Cowley et al., 2016* for a review), such as referencing, event-handling, filtering, dimensional reduction and artefact detection in channels, epochs or otherwise; all of which is time-consuming and therefore costly. Many of these operations require repeated human judgements, e.g. selection of artefactual independent components (ICs) (*Chaumon, Bishop & Busch, 2015*), leading to subjectivity, non-reproducibility of outcomes and non-uniformity of decisions. Nor is it possible that all such operations can ever be completely automated, as it is not possible to provide a ground-truth for computational methods by uniquely determination of the neural sources of EEG. With many relatively complex standard operations, code for EEG processing can also be harder to debug (*Widmann & Schröger, 2012*).

These issues illustrate the need for a software tool, a workflow management system, that helps to integrate the wealth of existing methods. Some standards have been suggested (*Keil et al., 2014*), however *Bigdely-Shamlo et al. (2015)* have pointed out that 'artefact removal and validation of processing approaches remain a long-standing open problem for EEG'. The EEGLAB toolbox (*Delorme & Makeig, 2004*) and its various plug-ins provide a wealth of functions, but in this ecosystem it remains difficult and time-consuming to build the necessary infrastructure to manage, regularise and streamline EEG preprocessing.

A workflow management system for data-processing pipelines helps to ensure that the researcher/analyst saves most of their cognitive effort for choosing analysis steps (not implementing them) and assessing their outcome (not debugging them). A regularised workflow maximises the degree to which each file is treated the same—for EEG this means to minimise drift in file-wise subjective judgements, such as estimating the accuracy of artefact detection algorithm(s) by visual inspection. A streamlined workflow can be enabled by separating the building of functions (for analysis or data management) from exploring and tuning the data. These features improve reproducibility and separate the menial from the important tasks. To meet these needs, in this paper we present the computational testing for automated preprocessing (CTAP) toolbox.

## Approach

The CTAP toolbox is available as a GitHub repository at https://github.com/bwrc/ctap. It is built on Matlab (R2015a and higher) and EEGLAB v13.4.4b; limited functions, especially non-graphical, may work on older versions but are untested.

The aim of CTAP is to regularise and streamline EEG preprocessing in the EEGLAB ecosystem. In practice, the CTAP toolbox extends EEGLAB to provide functionality for: (i) batch-processing using scripted EEGLAB-compatible functions; (ii) testing and

comparison of preprocessing methods based on extensive quality control outputs. The key benefits include:

- Ability to run a subset of a larger analysis
- Bookkeeping of intermediate result files
- Error handling
- Visualisations of the effects of analysis steps
- Simple to customise and extend
- Reusable code
- Feature and raw data export

We will next briefly motivate each of the benefits above.

**Incomplete runs:** A frequent task is to make a partial run of a larger analysis. This happens, for example, when new data arrives or when the analysis fails for a few measurements. The incomplete run might involve a subset of (a) subjects, (b) measurements, (c) analysis branches, (d) collections of analysis steps, (e) single steps; or any combination of these. CTAP provides tools to make these partial runs while keeping track of the intermediate saves.

**Bookkeeping:** A given EEG analysis workflow can have several steps, branches to explore alternatives and a frequent need to reorganise analysis steps or to add additional steps in between. Combined with incomplete runs, these requirements call for a system that can find the correct input file based on step order alone. CTAP does this and saves researchers time and energy for more productive tasks.

**Error handling:** Frequently, simple coding errors or abnormal measurements can cause a long batch run to fail midway. CTAP catches such errors, saves their content into log files for later reference and continues the batch run. For debugging purposes it is also possible to override this behaviour and use Matlab's built-in debugging tools to solve the issue.

**Visualisations:** It is always good practice to check how the analysis alters the data. CTAP provides several basic visualisations for this task giving the user additional insight into what is going on. See 'Results' for examples.

**Customisation:** In research it is vital to be able to customise and extend the tools in use. Extending CTAP with custom functions is easy as the interface that `CTAP_*.m` functions must implement is simple. Intermediate results are stored in EEGLAB format and can be directly opened with the EEGLAB graphical user interface (GUI) for inspection or manual processing.

**Code reuse:** The `CTAP_*.m` functions act as wrappers that make it possible to combine methods to build analysis workflows. Most analysis steps are actually implemented as standalone functions, such that they can be used also outside CTAP. In contrast to EEGLAB, CTAP functions do not pop-up configuration windows that interfere with automated workflows.

**Export facilities:** Exporting results might prove time-consuming in Matlab as there are no high-level tools to work with mixed text and numeric data. To this end, CTAP provides its own format of storing data and several export options. Small datasets can be

exported as, e.g. comma delimited text (csv) while larger sets are more practically saved in an SQLite database. CTAP also offers the possibility to store single-trial and average event-related potential (ERP) data in HDF5 format, which makes the export to R and Python simple.

In summary, CTAP lets the user focus on content, instead of time-consuming implementation of foundation functionality. In the rest of the paper, we describe how CTAP toolbox does this using a synthetic dataset as a running example.

We start with related work followed by the 'Materials and Methods' section detailing the architecture and usage of CTAP. The 'Results' section then describes the technical details and outcomes of a motivating example application. In the 'Discussion' section we set out the philosophy and possible uses of CTAP toolbox, including development as well as preprocessing; and describe issues and potential directions for future work.

## Related work

Many methods are available from the literature to facilitate automated preprocessing (*Agapov et al., 2016*; *Baillet et al., 2010*; *Barua & Begum, 2014*), and the rate of new contributions is also high.[1] In a milestone special issue (*Baillet, Friston & Oostenveld, 2011*) gathered many of the academic contributions available at that time. This special issue is quite skewed towards tools for feature extraction, which illustrates again the need for better/more up-to-date solutions for the fundamental stages of EEG processing.

Among tools dedicated to EEG processing, EEGLAB stands out for its large user community and high number of third-party contributors, to the degree that it is considered by some to be a de facto standard. Although EEGLAB functions can be called from the command-line interface and thus built into a preprocessing pipeline by the user's own scripts, in practice this is a non-trivial error-prone task.

Other popular tools focus on a more diverse set of signals, especially including magnetoencephalography (MEG). Brainstorm (*Tadel et al., 2011*), Fieldtrip (*Oostenveld et al., 2011*) and EMEGS (ElectroMagnetic EncaphaloGraphy Software) (*Peyk, De Cesarei & Junghöfer, 2011*) are all open source tools for EEG and MEG data analysis. Brainstorm in particular, but also the others, have originated with an emphasis on cortical source estimation techniques and their integration with anatomical data. Like EEGLAB, these tools are all *free* and *open source*, but based on the commercial platform Matlab (Natick, MA, USA), which can be a limitation in some contexts due to high licence cost. The most notable commercial tool is BrainVISION Analyzer (Brain Products GmbH, Munich, Germany), a graphical programming interface with a large number of features.

Tools which are completely *free* and *open source* are fewer in number and have received much less supplemental input from third parties. Python tools include MNE-Python for processing MEG and EEG data (*Gramfort et al., 2013*) and PyEEG (*Bao, Liu & Zhang, 2011*), a module for EEG feature extraction. MNE, like Brainstorm and Fieldtrip, is primarily aimed at integrating EEG and MEG data. Several packages exist for the R computing environment, e.g. (*Tremblay & Newman, 2015*), however these do not seem to be intended as general-purpose tools.

[1] For example, we conducted a search of the SCOPUS database for articles published after 1999, with "EEG" and "electroencephalography" in the title, abstract or keywords, plus "Signal Processing" or "Signal Processing, Computer-Assisted" in keywords, and restricted to subject areas "Neuroscience", "Engineering" or "Computer Science". The search returned over 300 hits, growing year-by-year from 5 in 2000 up to a mean value of 36 between 2010 and 2015.

[2] Also NeuroPype, a commercial Python-based graphical programming environment for physiological signal processing. However, to the authors' knowledge, it has not been documented in a peer reviewed publication.

However, CTAP was designed to complement the existing EEGLAB ecosystem, not to provide a stand-alone preprocessing tool. This is an important distinction, because there exist some excellent stand-alone tools which work across data formats and platforms (*Bellec et al., 2012*; *Ovaska et al., 2010*)[2]; these features are valuable when collaborators are trying to work across, e.g. Windows and Linux, Matlab and Python. However, we do not see a need in this domain; rather we see a need in the much narrower focus on improving the command-line interface batch-processing capabilities of EEGLAB.

We have chosen to extend EEGLAB because it has received many contributions to the core functionality, and is thus compatible with a good portion of the methods of EEG processing from the literature. Some compatible tools from the creators of EEGLAB at the Swartz Centre for Computational Neuroscience (SCCN) are detailed in (*Delorme et al., 2011*), including tools for forward head modelling, estimating source connectivity and online signal processing. Other key third-party preprocessing contributions to EEGLAB include SASICA (*Chaumon, Bishop & Busch, 2015*), FASTER (*Nolan, Whelan & Reilly, 2010*) and ADJUST (*Mognon et al., 2011*), all semi-automated solutions for selection of artefactual data.

In terms of similar tools *Bigdely-Shamlo et al. (2015)* released the PREP pipeline for Matlab, which also uses the EEGLAB data structure. PREP introduces specific important functionality for referencing the data, line noise removal and detecting bad channels. PREP is aimed only at experiment-induced artefacts and not those deriving from subject-activity such as, e.g. blinks, and is designed to be complementary to the various algorithm toolboxes for artefact-removal by focusing on early-stage processing. In similar vein, CTAP is intended to be complementary to existing toolboxes *including PREP*.

For example, methods from FASTER and ADJUST are featured in CTAP as options for detecting bad data. This integration of existing solutions illustrates one core principle of CTAP: it aims to extend an existing rich ecosystem of EEG-specific methods, by meeting a clear need within that ecosystem for a workflow management system. The ready-made automation of batching and bookkeeping gives the user a distinct advantage over the common approach of 'EEGLAB + a few scripts', which seems simple on its face, but in practice is non-trivial as the number and complexity of operations grows. As all algorithms added to CTAP will produce quality control outputs automatically, fast performance comparison is possible between methods or method parameters, speeding the discovery of (locally) optimal solutions. The system has potential to enable such parameter optimisation by automated methods, although this is not yet implemented.

## MATERIALS AND METHODS

The core activity of CTAP is preprocessing EEG data by cleaning artefacts, i.e. detection and either correction or removal of data that is not likely to be attributable to neural sources. CTAP is able to operate on three different temporal granularities: channel, epoch and segment. Channel operations affect the entire time series at one spatial location. Epoch operations are performed on one or several epochs produced by EEGLAB epoching function. Finally, segments are fixed time-windows around specific events which can be extracted from both channel and epoch levels, see Fig. 1. An example of a typical segment

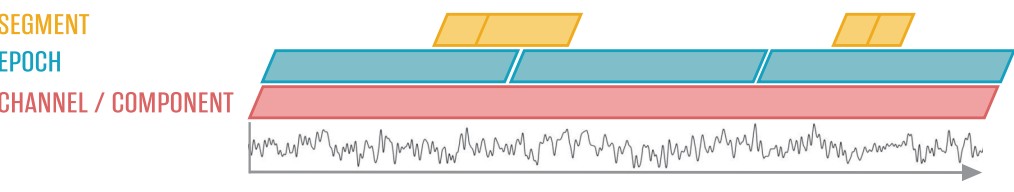

**Figure 1** Relationship of the time domain data constructs dealt with in CTAP.

could be a blink artefact with a window wide enough to include the entire blink waveform. Further functionality is provided for independent component analysis (ICA)-based methods. Artefact-detection methods based on some flavour of ICA algorithm have been shown to outperform temporal approaches (*Delorme, Sejnowski & Makeig, 2007*). It was also shown that ICs are valid representations of neural sources (*Delorme et al., 2012*). CTAP can thus help to combine the existing methods for EEG signal processing.

## Outline of usage

Figure 2 shows the core components of CTAP. The coloured boxes represent entities that the user has to specify in order to use CTAP. These are:

- What analysis functions to apply and in which order (analysis pipe)
- Analysis environment and parameters for the analysis functions (configuration)
- Which EEG measurements/files to process (measurement configuration)

Typically, the analysis is run by calling a single script that defines all of the above and passes these on to the `CTAP_pipeline_looper.m` function, that performs all requested analysis steps on all specified measurements. In the following, we describe in more detail how the configurations are made, how the pipe is executed, what outputs it provides and what options the user has to control the pipe. The complete details of all these aspects of CTAP are provided in the wiki pages of the GitHub repository, which will be referenced below as 'the wiki'.[3]

[3] https://github.com/bwrc/ctap/wiki.

### *Configuration*

In CTAP a large analysis is broken down into a hierarchical set of smaller entities: steps, step sets, pipes and branches. Several analysis *steps* form a *step set* and an ordered sequence of step sets is called a *pipe*. Pipes can further be chained to form *branches*. The smallest unit is the analysis step which might be e.g. a filtering or a bad channel detection operation. A step is represented by a single call to a `CTAP_*.m`-function. Step sets and pipes are used to chop the analysis down into smaller chunks that are easy to move around if needed.

Intermediate saves are performed after each step set and therefore the organisation of steps into step sets also affects the way the pipe shows up on disk. Intermediate saves provide a possibility run the whole analysis in smaller chunks and to manually check the mid-way results as often needed, e.g. while debugging. Further on, the ability to create branches is important to help explore alternative ways of analysing the same data.

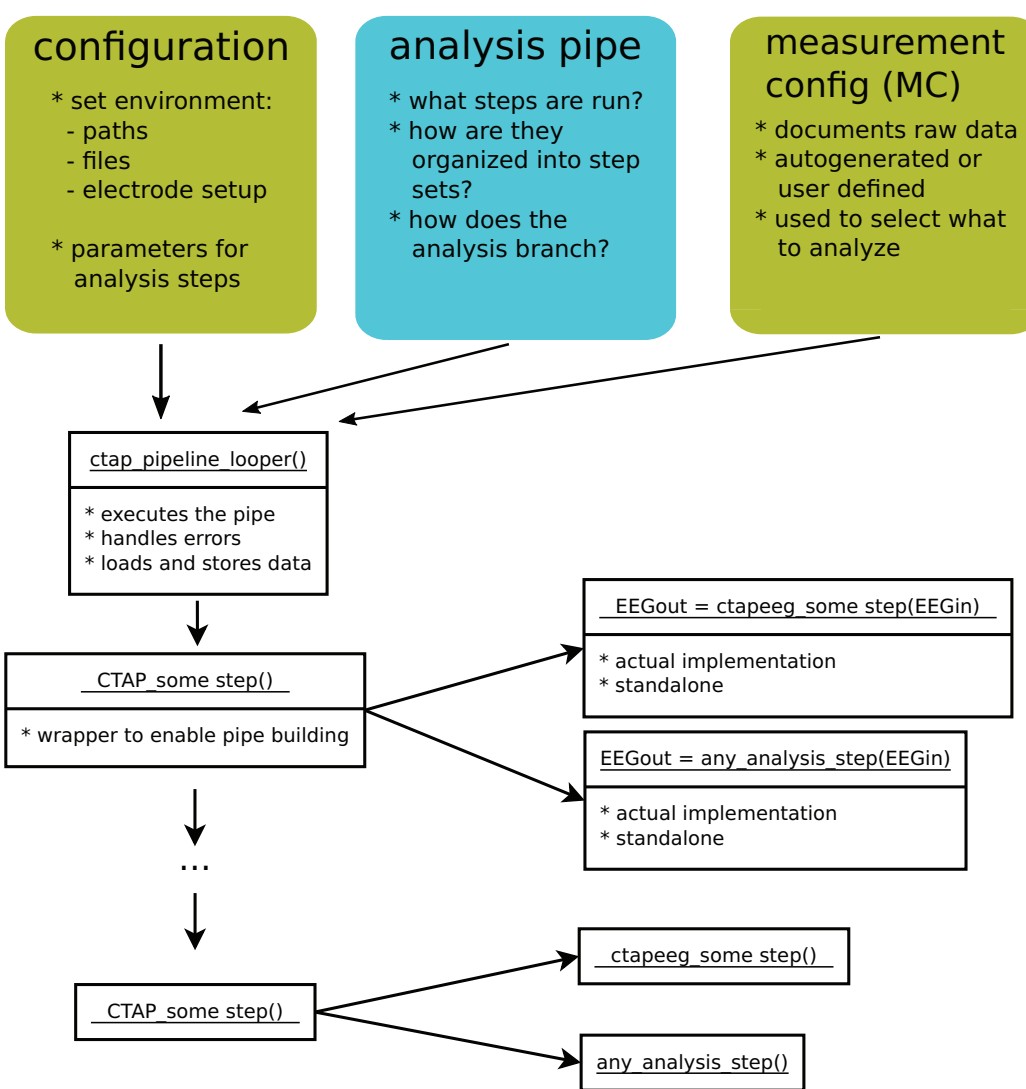

**Figure 2 An overview of the core logic of CTAP.** 'Configuration', 'analysis pipe' and 'measurement config' illustrate the parts that a user must specify. White boxes represent Matlab functions, with the function-name on top.

To specify the order of steps and sets within a pipe, we recommend to create a single m-file for each intended pipe.[4] This file will define both the step sets as well as all the custom parameters to be used in the steps. Default parameters are provided, but it is optimal to fine tune the behaviour by providing one's own parameters. Both pipe and parameter information is handled using data structures, rather than hard-coding. CTAP then handles assignment of parameters to functions based on name matching.

Once the steps and their parameters are defined, the last requirement to run the pipe is to define the input data. In CTAP the input data are specified using a table-like structure called *measurement config* that lists all available measurements, the corresponding raw EEG files, etc. This dedicated measurement config data structure allows for an easy selection of what should be analysed and it also helps to document the project. It can

[4] For an example, see the `cfg_manu.m` in the repository.

be created manually or auto-generated based on a list of files or a directory. The former allows for full control and enforces project documentation whereas the latter is intended for effortless one-off analyses. Both spreadsheet and SQLite formats are supported.

In the last required step before pipeline execution, the configuration struct and the parameter struct are checked, finalised and integrated by `cfg_ctap_functions.m`.

### Pipe execution

Once all prerequisites listed above have been specified, the core `CTAP_pipeline_looper.m` function is called to run the pipe. This function takes care of loading the correct (initial or intermediate) data set, applying the specified functions from each step set, and intermediate saving of the data. The looper manages error handling such that it is robust to crashing (unless in Debug mode), and will simply skip the remaining steps for a crashed file. Other settings determine how to handle crashed files at later runs of the pipe (see Documentation).

`CTAP_pipeline_looper.m` is designed to accept functions named `CTAP_*.m`, as these are defined to have a fixed interface. They take two arguments: data (EEG) and configuration struct (Cfg); and they return the same after any operations. Some `CTAP_*.m` perform all operations (e.g. call EEGLAB functions) directly, while others call a corresponding `ctapeeg_*.m` function that actually implements the task. Hence `CTAP_*.m` functions can be regarded as wrappers that facilitate batch-processing by providing a uniform interface. They also implement, e.g. the plotting of quality control figures. Since `CTAP_*.m` functions are quite simple, new ones can easily be added by the user to include new analysis steps, working from the provided `CTAP_template_function.m`. Users can also call the `ctapeeg_*.m` functions directly as part of their own custom scripts, since these are meant to be used like any EEGLAB analysis function.

Analysis results are saved separately for each pipe. A typical structure contains:

- Intermediate results as EEGLAB datasets, in one directory per step set; names are taken from the step set IDs as defined by the user, prefixed by step number.
- `export` directory contains exported feature data (txt, csv or SQLite format).
- `features` directory: computed EEG features in Matlab format.
- `logs` directory: log files from each run.
- `quality_control` directory: quality control plots, reflecting the visualisations of analysis steps chosen by the user.

Apart from running the complete pipe at once the user has many options to run just a subset of the pipe, analyse only certain measurements or otherwise adjust usage. Table 1 gives some examples.

## Analytic methods

As presented, CTAP is primarily a framework for analysis management; however it contains a number of analysis functions, functions for evaluation and data-management functions including a way to generate synthetic datasets for testing (for details see function documentation). The user is easily able to add their preferred functions, but may note

**Table 1 Some advanced ways to use the pipe.**

| Usage options | Possible reasons | How |
|---|---|---|
| Subset step sets | Investigate a bug; recompute only intermediate results | Set run sets to subset index, e.g. `Cfg.pipe.runSets = 3:5` |
| Run *test* step set | Test new feature before including in pipe | Add step set with id 'test', then set `Cfg.pipe.runSets = 'test'` |
| 'Rewire' the pipe | Test an alternative ordering of existing steps or temporarily change the input of some step | Set the `.srcID` of a given step set equal to the id of another |
| Measurement configuration filter | Run pipe for: subset of test subjects, or: measurements classes with separate configurations, e.g. pilots | Use function `struct_filter.m` |
| Run in debug mode | Develop new method in CTAP | Set `CTAP_pipeline_looper` parameter 'debug', `true` |
| Overwrite obsolete results | Update part of pipe: write new step set output over existing files | Set `CTAP_pipeline_looper` parameter 'overwrite', `true` |
| Write files from failed step sets | Check partial outcome of step set | Set `CTAP_pipeline_looper.m` parameter 'trackfail', `true` |
| Turn off intermediate saves | Extract numerical/visual analytics without producing updated files | Set `stepSet(x).save = false;` set `stepSet(x+1).srcID = stepSet(x−1).id` |

the available functions as a quick way to start. All provided functions, for analysis, evaluation or data-handling, have default parameters which may serve as a starting point.

Almost all EEG processing methods in CTAP are either novel or rewritten from original source, usually because of the unintended side-effects of the original code, such as graphical pop-ups. Thus the outputs are similar to those of original EEGLAB or other toolbox methods, but the code base is refactored.

The highlights of available `CTAP_*.m` functions include:

- Functions to load data (and extract non-EEG data, e.g. ECG), events (and modify them) or channel locations (and edit them);
- Functions to filter, subset select (by data indices or by events), re-reference, epoch or perform ICA on the data;
- Functions to detect artefactual data, in channels, epochs, segments or ICA components, including:

  - Variance (*channels*),
  - Amplitude threshold (*epochs, segments, ICA components*),
  - EEGLAB's channel spectra method (*channels, epochs*),
  - Metrics from the FASTER toolbox (*channels, epochs, ICA components*),
  - Metrics from the ADJUST toolbox (*ICA components*),
  - Additionally bad data can be marked by events where detection is performed by some external method;

- Functions to reject bad data, normalise or interpolate;
- Functions to extract time and frequency domain features, and create visualisations of data (as described below).

**Peer**J Computer Science

## Outputs

CTAP provides a number of novel outputs for evaluation and data management.

**Visual evaluation:** CTAP automatically produces plots that help the user to answer questions such as: what has been done, what the data looks like and was an analysis step successful or not. The following selected visualisations are illustrated in 'Results':

- Blinks: detection quality, blink ERP
- Bad segments: snippets of raw EEG showing detections
- EEG amplitudes: amplitude histograms, peeks
- Filtering: PSD comparison
- ICA: IC scalp-map contact sheets, zoom-ins of bad components

**Quantitative evaluation:** Every major pipe operation writes a record to the main log file. Data rejections, including channels, epochs, ICs or segments, are summarised here and also tabulated in a separate 'rejections' log. Values are given for how much data was marked as bad, and what percentage of the total was bad. If more than 10% of data is marked bad by a single detection, a warning is given in the main log. In addition, useful statistics of each channel are logged at every call to `CTAP_peek_data.m`, based on the output of the EEGLAB function `signalstat.m`. Data-points include trimmed and untrimmed versions of mean, median, standard deviation as well as skewness, kurtosis and normality testing. The set of statistics estimated for every data channel is saved in Matlab table format and also aggregated to a log file.

**Feature export:** Extracted EEG features are stored internally as Matlab structs that fully document all aspects of the data. These can be used to do statistical analysis inside Matlab. However, often users like to do feature processing in some other environment such as R or similar. For this, CTAP provides export functionality that transforms the EEG feature mat files into txt/csv text files, and/or an SQLite database. For small projects (e.g. up to 10 subjects and 16 channels) txt/csv export is feasible but for larger datasets SQLite is more practical.

## System evaluation

To showcase what CTAP can do we present in this paper the output of an example analysis using synthetic data. The example is part of the CTAP repository; methods are chosen to illustrate the range of possibilities in CTAP, rather than for the qualities of each method itself. Thus, for example, we include the CTAP-specific blink-correction method alongside simple amplitude thresholding, to exemplify different ways to handle artefacts.

### Toy data

CTAP provides a motivating example that can also be used as a starting point for one's own analysis pipe. The example is based on synthetically generated data with blink, myogenic (EMG) and channel variance artefacts to demonstrate the usage and output of CTAP. The example is part of the repository and the details of the synthetic data generation process are documented in the wiki.[5] Shortly, synthetic data is generated from

---

[5] https://github.com/bwrc/ctap/wiki/
syndata-generation.

6 http://bbci.de/competition/iv/desc_1.
html.

seed data using `generate_synthetic_data_manuscript.m`, which first converts the example dataset to EEGLAB-format and then adds artefacts to the data. Seed data included in the repository is from the BCI competition IV dataset 1[6], recorded with BrainAmp MR plus at 100 Hz on 59 channels. The generated 10 min dataset is sampled at 100 Hz and has 128 EEG channels, two mastoid channels and four EOG channels. It occupies ~32 MB on disk.

Artefacts added to the data include 100 blinks (generated by adding an exponential impulse of fixed duration, with amplitude that decreases linearly from front to rear of the scalp-map); and 50 periods of EMG (generated by adding a burst of noise across an arbitrary frequency band, at a high amplitude that decreases linearly away from a random centre-point). Also six channels are 'wrecked' by randomly perturbing the variance, either very high (simulating loose electrodes) or very low (simulating 'dead' electrodes).

### Analysis steps

An example pipeline, described in the CTAP repository in the file `cfg_manu.m`, is run on the synthetic data using `runctap_manu.m`. Here, we describe the non-trivial analysis steps in order of application. For each step, we first describe the method; then the 'Results' section shows the generated outcomes in terms of data quality control statistics and visualisations. The pipe below is shown to illustrate context of the steps, and is an abridged version of the repository code.

```
stepSet(1).id = '1_LOAD';
stepSet(1).funH = {@CTAP_load_data,...
                   @CTAP_load_chanlocs,...
                   @CTAP_reref_data,...
                   @CTAP_peek_data,...
                   @CTAP_blink2event};

stepSet(2).id = '2_FILTER_ICA';
stepSet(2).funH = {@CTAP_fir_filter,...
                   @CTAP_run_ica};

stepSet(3).id = '3_ARTIFACT_CORRECTION';
stepSet(3).funH = {@CTAP_detect_bad_comps,...
                   @CTAP_filter_blink_ica,...
                   @CTAP_detect_bad_channels,...
                   @CTAP_reject_data,...
                   @CTAP_interp_chan, ...
                   @CTAP_detect_bad_segments,...
                   @CTAP_reject_data,...
                   @CTAP_run_ica,...
                   @CTAP_peek_data};
```

**Before-and-after 'Peeks':** The `CTAP_peek_data.m` function is called near the start (after initial loading and re-referencing) and the end of the pipe. Visual inspection of raw data is a fundamental step in EEG evaluation and quantitative inspection of channel-wise statistics is also available. A logical approach is to compare raw data at same time-points from before and after any correction operations. If ICA-based corrections are made, the same approach can also be used on the raw IC data. `CTAP_peek_data.m` expedites this work, and thus helps to regularise data inspection and facilitate comparison.

`CTAP_peek_data.m` will generate raw data plots and statistics of a set of time-points (points are generated randomly by default or can be locked to existing events). These 'peek-points' are embedded as events which can then generate peeks at a later stage in the pipe, allowing true before-and-after comparisons even if the data time course changes (due to removal of segments). If no peek-point data remains at the after-stage, no comparison can be made; however (especially if peek-points are randomly chosen), such an outcome is itself a strong indication that the data is very bad, or the detection methods are too strict.

`CTAP_peek_data.m` includes plotting routines for signal amplitude histograms as well as for raw EEG data. Many EEG artefacts cause large changes in signal amplitudes, and consequently several basic, yet effective, EEG artefact detection methods are based on identifying samples exceeding a given amplitude threshold. On the other hand, even in controlled measurement conditions, individual baseline variation can affect the amplitude of the recorded signal. Hence, accurate knowledge of the average signal amplitude is often important.

**Blink detection:** The function `CTAP_blink2event.m` is called early in the pipe to mark blinks. It creates a set of new events with latencies and durations matched to the detected blinks. The current blink detection implementation is based on a modified version of the EOGERT algorithm by *Toivanen, Pettersson & Lukander (2015)*.[7] The algorithm finds all local peaks in the data, constructs a criterion measure and classifies peaks into blinks and non-blinks based on this measure.

**Filtering:** CTAP filtering produces plots of filter output and tests of functionality as standard. `CTAP_fir_filter.m` uses the firfilt-plug-in[8] to do filtering, as it replaces the deprecated function `pop_eegfilt.m` and provides more sensible defaults. Version 1.6.1 of firfilt ships with EEGLAB. Other CTAP-supported filtering options are described in documentation.

**Blink removal:** Blinks can either be rejected or corrected. We showcase correction using a method that combines blink-template matching and FIR high-pass filtering of blink-related ICs following ideas presented by *Lindsen & Bhattacharya (2010)*. The method is not part of EEGLAB, but an add-on provided by CTAP.[9]

*Bad ICA component detection* is performed by first creating ICs with `CTAP_run_ica.m`[10], and then using one of several options from `CTAP_detect_bad_comps.m` to detect artefactual ICs. The blink template option compares mean activity of detected blink events to activations for each IC.

`CTAP_filter_blink_ica.m` is used to *filter blink-related IC data*, and reconstruct the EEG using the cleaned components. The success of the blink correction is evaluated

---

[7] See code repository at https://github.com/bwrc/eogert.

[8] https://github.com/widmann/firfilt.

[9] Including all parts described above, this particular blink-correction method is unique to CTAP.

[10] Default algorithm is FastICA, requiring the associated toolbox on the user's Matlab path.

using blink evoked response potentials (ERPs) which are simply ERPs computed for blink events (see e.g. *Frank & Frishkoff, 2007* for details).

**Detect raw-data artefacts:** *Bad channels* were detected based on channel variance, with the function `vari_bad_chans.m`. Log relative variance $\chi$ was computed for all channels using the formula $\chi = \log(\frac{channel\ variance}{\text{median}(channel\ variance)})$. Values of $\chi$ more than three median absolute deviations away from median ($\chi$)were interpreted as deviant and labelled as bad.

For *bad segments*, i.e. short segments of bad data over multiple channels, a common approach (in e.g., EEGLAB) is analysis of fixed length epochs, which is good for ERP experiments. Alternatively for working with continuous data, CTAP also provides the option of amplitude histogram thresholding. Many types of large artefacts can be easily found using simple histogram-based thresholding: a predefined proportion of most extreme amplitude values are marked as artefacts and segments are expanded around these. This can improve, e.g. ICA analysis of low density EEG by freeing ICs to capture neural source signals.

For all `CTAP_detect_bad_*.m` functions, for whichever detection method option is used (user-defined options are also straightforward to add), a field is created in the EEG struct to store the results. Another field collects pointers to all results detected before a rejection. This logic allows the user to call one or many detection functions, possibly pooling the results of several approaches to bad data detection, and then pass the aggregate results to the `CTAP_reject_data.m` function.

**Rejection:** CTAP usage logic suggests that one or more detect operations for a given data type, e.g. channels, *or* epochs, *or* components, should be followed by a reject operation. It is bad practice to detect bad data across modalities, e.g. channels and epochs, before rejecting any of it, because artefacts of one type may affect the other. `CTAP_reject_data.m` checks the detect field to determine which data type is due for rejection, unless explicitly instructed otherwise. Based on the data labelled by prior calls to detection functions, `CTAP_reject_data.m` will call an EEGLAB function such as `pop_select.m` to remove the bad data. Upon rejection, visualisation tools described are used to produce plots that characterise the rejected components.

Note that data *rejection* is only necessary if there exists no method to *correct* the data, e.g. as is provided for the CTAP blink removal method. In that case the call to the `CTAP_detect_bad_*.m` function is not followed by a call to `CTAP_reject_data.m`, because the method *corrects* the artefactual ICs rather than simply deleting them.

**After peek:** Finally, the `CTAP_peek_data.m` function is called again, providing comparator data at the same points as the initial peek call. A useful approach is to call `CTAP_run_ica.m` again *after* all artefact correction steps. The resulting set of raw IC activations can be plotted by calling `CTAP_peek_data.m`, and a careful examination should reveal the presence or absence of any remaining sufficiently large artefacts. This is a convenient way to, for example, determine whether the blink detection has identified all blink ICs.

## RESULTS

In this section, we show the output of CTAP as applied to the synthetic dataset, based on the analysis-pipe steps shown above. The pipe outputs ~30 MB of EEG data after each step set, thus after debugging all steps can be expressed as one set, and data will occupy

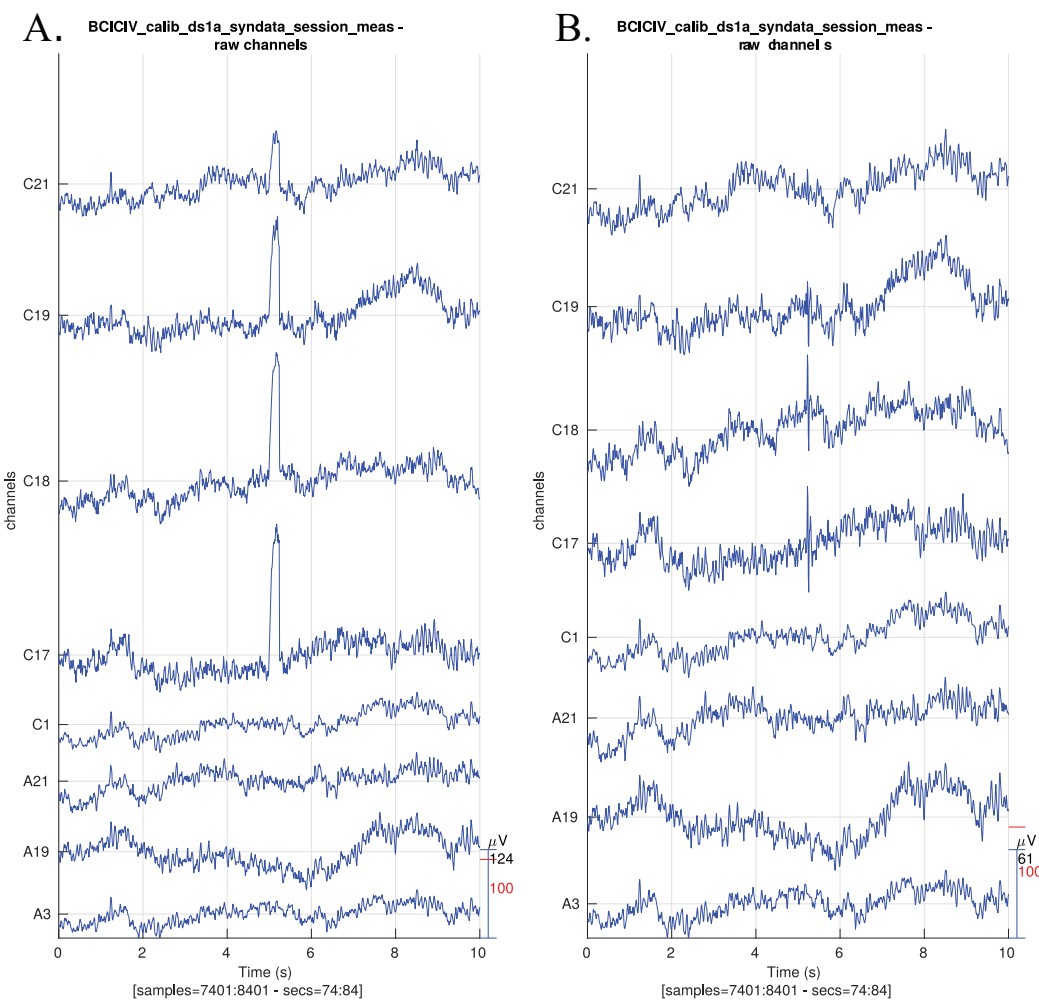

**Figure 3 Raw EEG data centred around a synthetic blink (A) before preprocessing and (B) after preprocessing.** The blinks have been largely removed and the EEG activity around blinks has remained intact. Note that the *y*-axis scales differ slightly.

~62 MB (before and after processing). Additionally, the quality control outputs of this pipe occupy ~70 MB of space, mostly in the many images of the peek-data and reject-data functions.

## Before-and-after 'Peeks'

**Raw data:** Figure 3 shows raw data before and after preprocessing.

**EEG amplitudes:** The signal amplitude histograms of a sample of good and bad channels from the synthetic data set are shown in Fig. 4. This can be useful for finding a suitable threshold for bad segment detection, or e.g. to detect loose electrodes. The post-processing plots show the improvement in channel normality.

**Statistical comparison:** Some of the first-order statistics calculated for before-and-after comparisons are plotted in Fig. 5, averaged over all channels. This method allows inspection of global change in the signal, which overall can be expected to become less broad (smaller range) and less variable (smaller SD) after cleaning of artefacts.

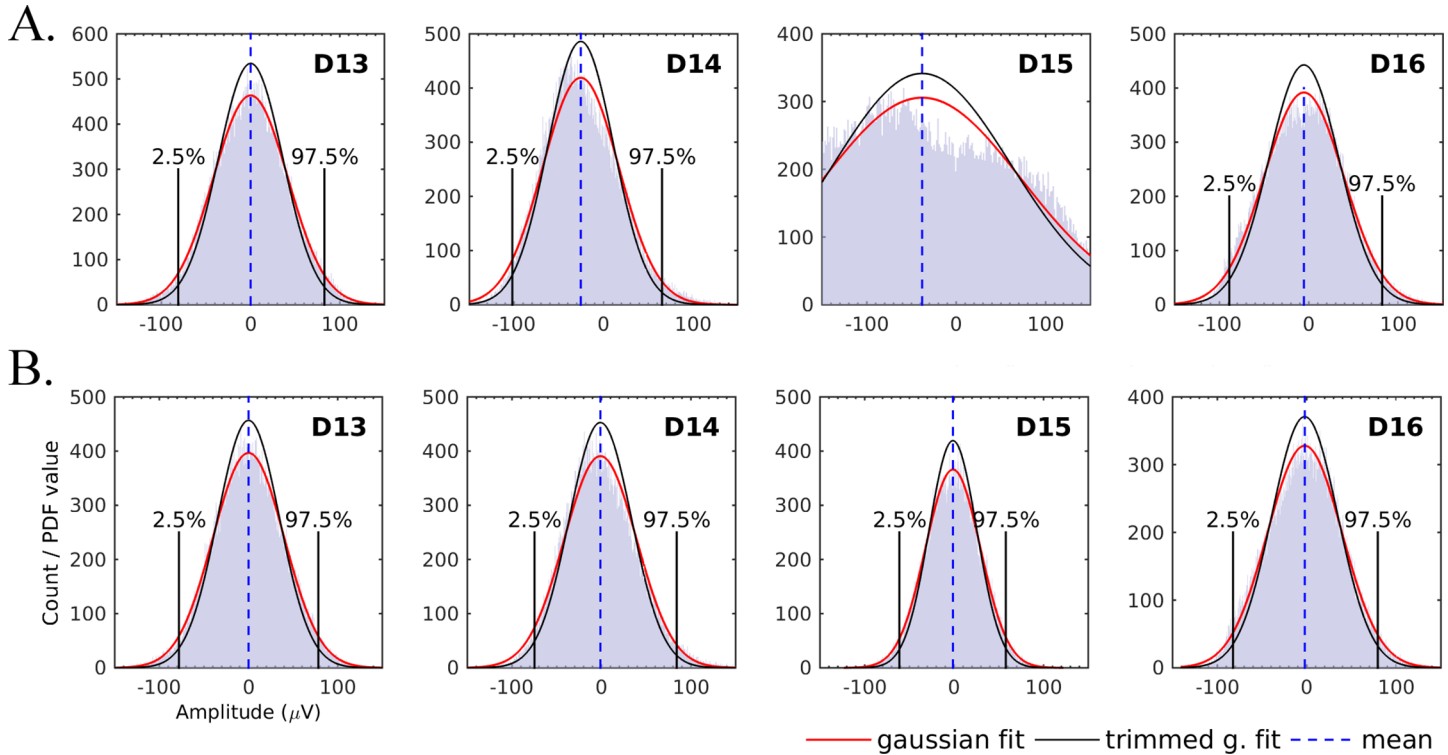

**Figure 4** **EEG amplitude histograms for four channels (A) before preprocessing and (B) after preprocessing.** Fitted normal probability density function (PDF) is shown as red solid curve. Upper and lower 2.5% quantiles are vertical black solid lines; data inside these limits was used to estimate the trimmed standard deviation (SD) and normal PDF fitted using trimmed SD is shown as black solid curve. Distribution mean is vertical dashed blue line. Channel D15 has clearly been detected as bad, removed and interpolated.

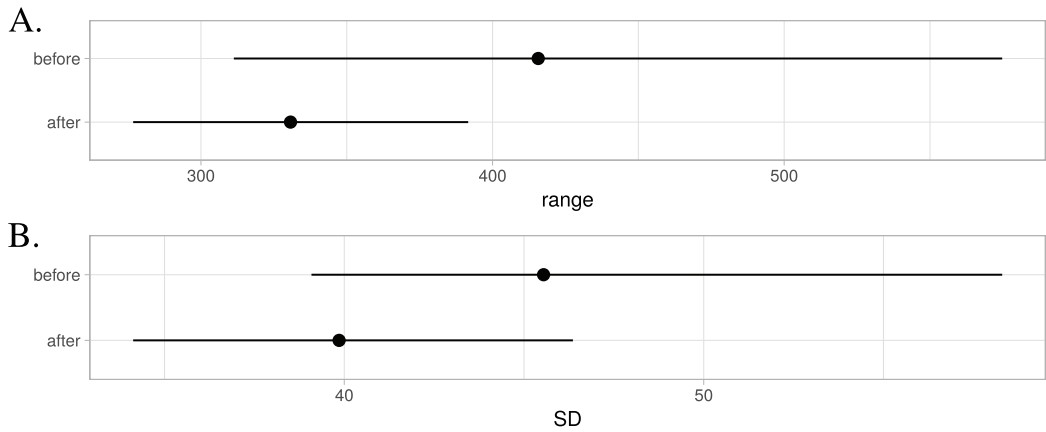

**Figure 5** **Changes in channel statistics for range (A) and standard deviation (SD) (B).** Mean over channels is indicated using a dot and the range spans from 5th to 95th percentile.

## Blink detection

The EOGERT blink detection process visualises the classification result for quality control purposes, as shown in Fig. 6. Such figures make it easy to spot possible misclassifications. In our example, all 100 blinks inserted into the synthetic data were detected.

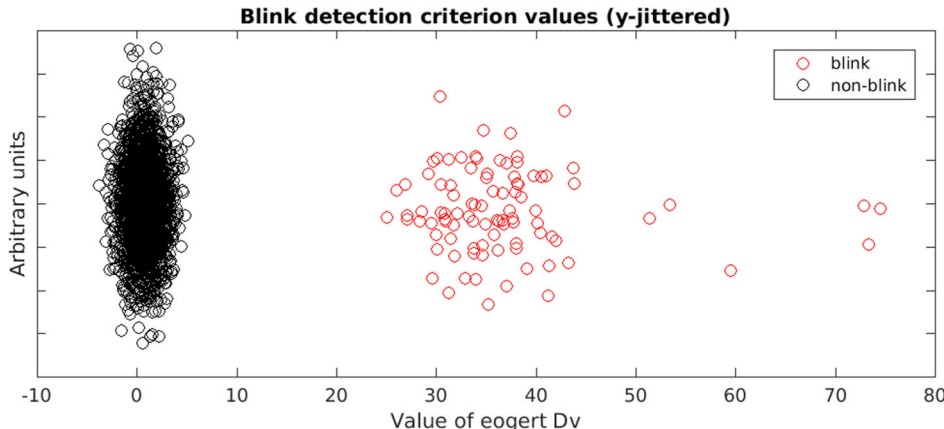

**Figure 6 Scatter plot of the criterion used to detect blinks.** Horizontal-axis shows the criterion value while vertical-axis is random data to avoid over-plotting. The classification is done by fitting two Gaussian distributions using the EM algorithm and assigning labels based on likelihoods.

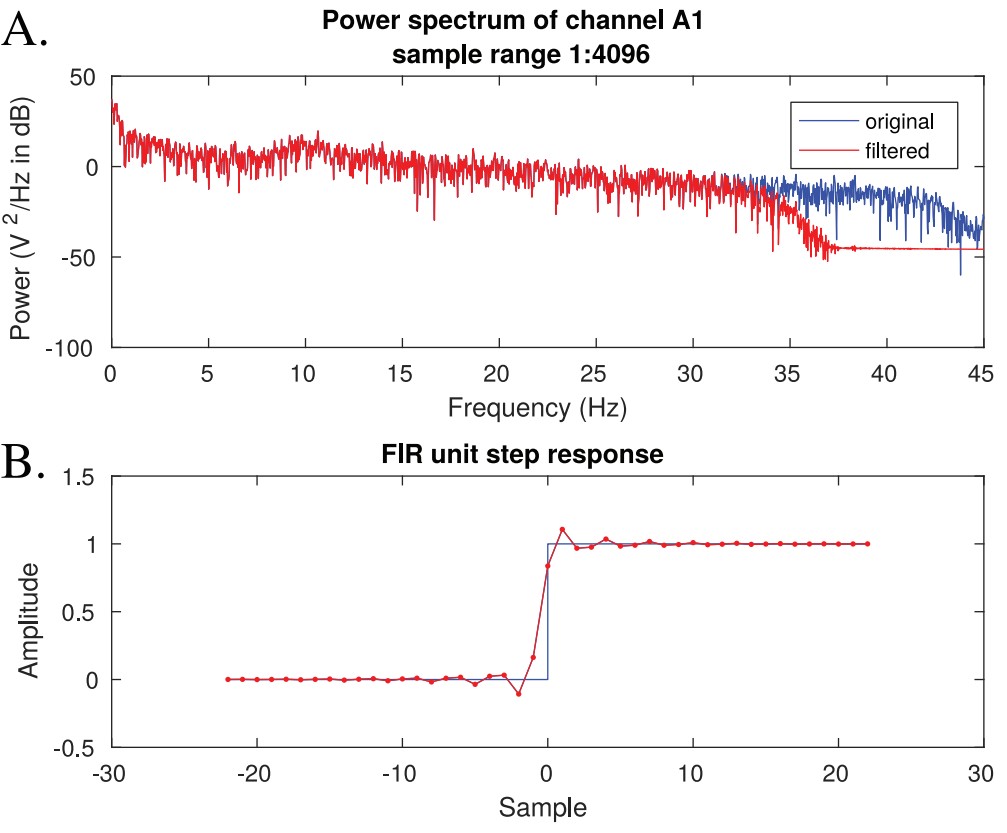

**Figure 7 A visual of filtering effects.** (A) The effects of filtering on power spectrum, (B) the filter's unit step response which can be used to assess, e.g. the filter's effect on ERP timings.

## Filtering

Figure 7 shows one of the outputs of the FIR filtering. This figure can be used to check that the filter has the desired effect on power spectrum and that its response to a unit step function is reasonable.

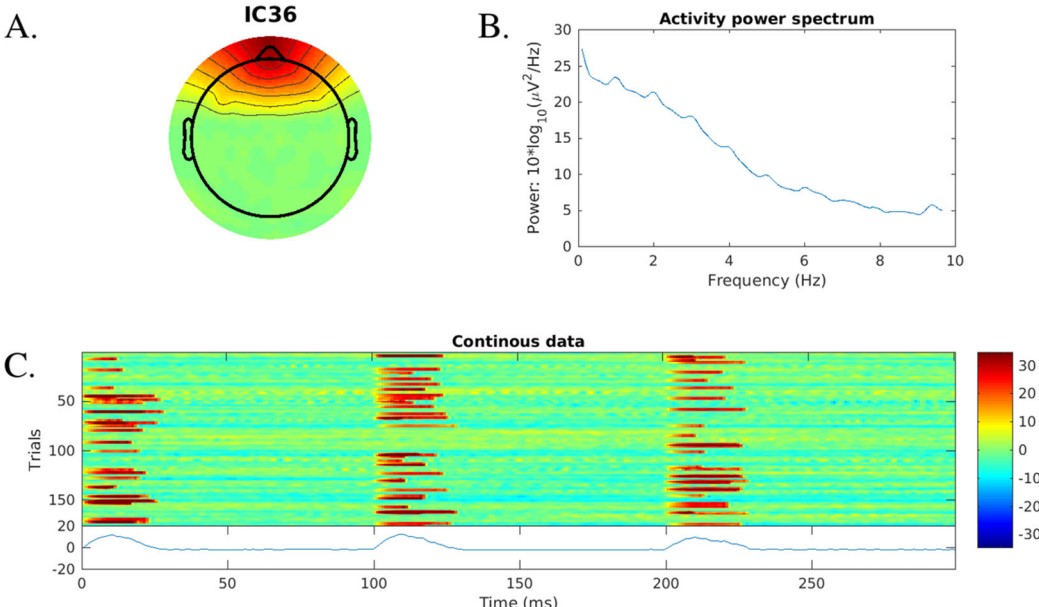

**Figure 8 Independent component information plot for a blink-related ICA component found using blink template matching.** Shown are (A) component scalp map, (B) power spectrum and (C) a stacked plot of the time series (using `erpimage.m`). (C) Shows only 200 first 300 ms segments of the data. The synthetic blinks start at full seconds by design.

## Blink removal

**Bad ICA component detection:** An example of a plot for ICA rejection is given in Fig. 8, showing some basic properties of a blink-related IC.

**Filter blink IC data:** The ERP-evaluated success of the blink correction is shown in Fig. 9. The correction method clearly removes most of the blink activity. As blink related ICs are corrected instead of rejected, effects on the underlying EEG are smaller. The result may have some remainder artefact (e.g. visible in Fig. 3 as small spikes after 5 s in channels C17, C18), which may motivate the complete removal of blink-related ICs instead of filtering.

## Detect and reject raw-data artefacts

**Bad channels:** In total 10 bad channels were found which included all six 'wrecked' channels—this shows the algorithm is slightly greedy, which is probably preferable in the case of a high-resolution electrode set with over 100 channels. Bad channels are rejected and interpolated before proceeding (not plotted as it is a straightforward operation).

**Bad segments:** An example of bad segment detection, using simple histogram-based amplitude thresholding, is shown in Fig. 10. In this case, the bad data is high amplitude EMG but in a general setting, e.g. motion artefacts often exhibit extreme amplitudes. Using these figures the user can quickly check what kind of activity exceeds the amplitude threshold in the dataset.

Of the 50 EMG artefacts inserted in the synthetic data, 37 still existed at least partially, at the end of pipe. The low rejection percentage is due to the fact that EMG is more of a

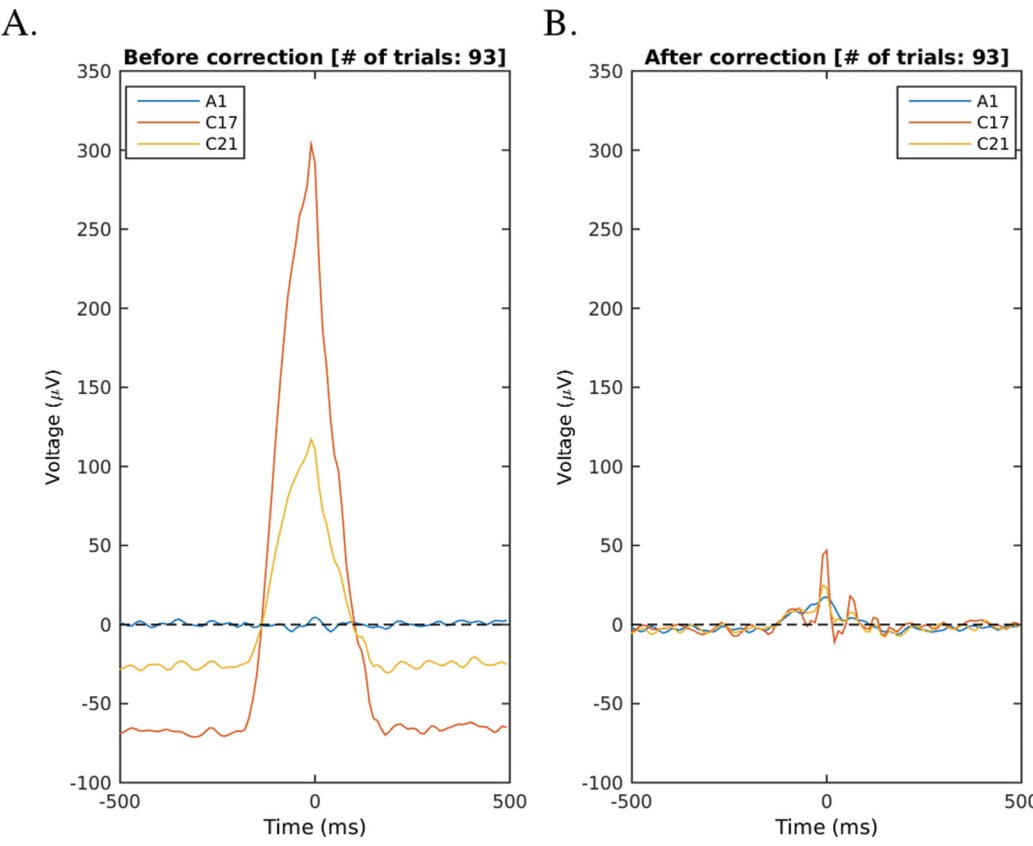

**Figure 9** **An example of the blink ERP.** (A) The blink-centred ERP before correction with a clearly visible blink signal. (B) The same plot after correction. The blink is clearly removed but the underlying EEG remains largely unaffected because the correction was done in IC base. Channel C17 shows highest blink amplitudes in the synthetic dataset.

change in frequency spectrum than in amplitude, yet the pipe looked for deviant amplitudes only.

## After peek

The data comparisons after various artefact removal operations, Figs. 3 and 4, illustrate the success or failure of the pipe. Of course there are a large number of permutations for how this can be done—it is the CTAP philosophy to facilitate free choice among these options, with the least implementation overhead. Additionally, the final plots of raw IC activations should show if there remains any artefacts in the data. For example, Fig. 11 shows a segment of raw data for the first 1/3 of ICs for the synthetic dataset, with clear indications of issues remaining in the data.

## DISCUSSION

We have presented CTAP, an EEG preprocessing workflow-management system that provides extensive functionality for quickly building configurable, comparative, exploratory analysis pipes. Already by shifting the researcher's focus from scripting to analysis, CTAP can help reduce human effort, subjectivity and consequent error.

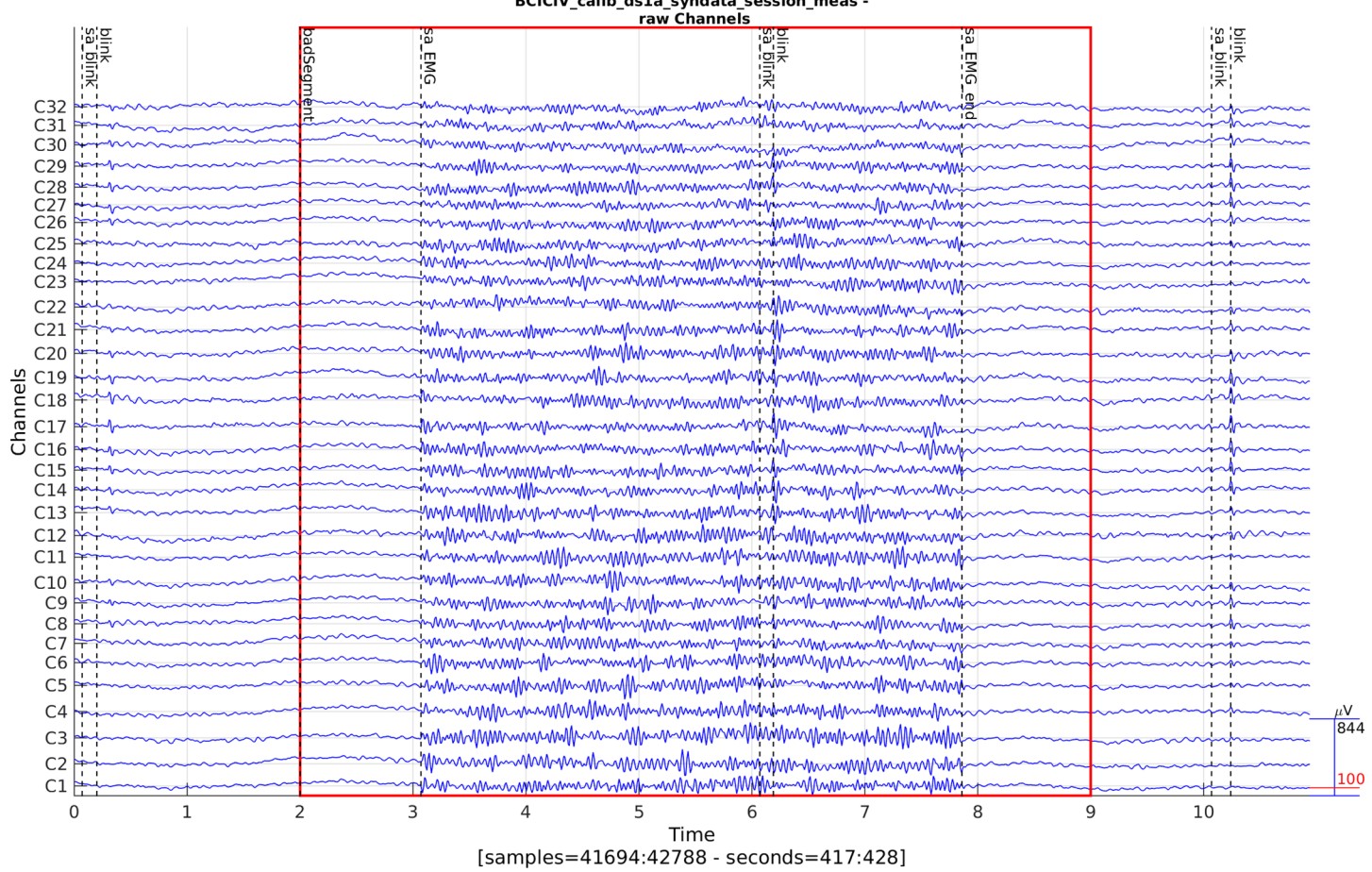

**Figure 10 A bad data segment detected by histogram-based amplitude thresholding** (`CTAP_detect_bad_segments.m`)**.** The 32-channel subset closest to the forehead is shown (C1–C32). The red lines mark the area of the bad segment.

Specifically, the system can reduce the work load of the user by streamlining analysis specification away from function coding. It can improve reliability and objectivity of the analysis by helping users treat each file in a dataset in a uniform, regular manner. CTAP output can also be more easily reproduced because manual processing steps have been minimised. This enables the user to perform multiple comparative analyses for testing the robustness of the results against different preprocessing methods.

## Philosophy, benefits and issues

CTAP provides many default parameters, and streamlines many features into a handful of wrapper functions. This is in order to facilitate rapid build and testing of analysis pipes. The philosophy is to prevent users becoming stuck in a single approach to the data because they have invested time in building the preprocessing code for it from scratch; or worse, because they have completed a laborious manual processing task and cannot afford to repeat it.

Computational testing for automated preprocessing structures pipes in *function, argument* specification files. This approach, instead of only making scripts that call the

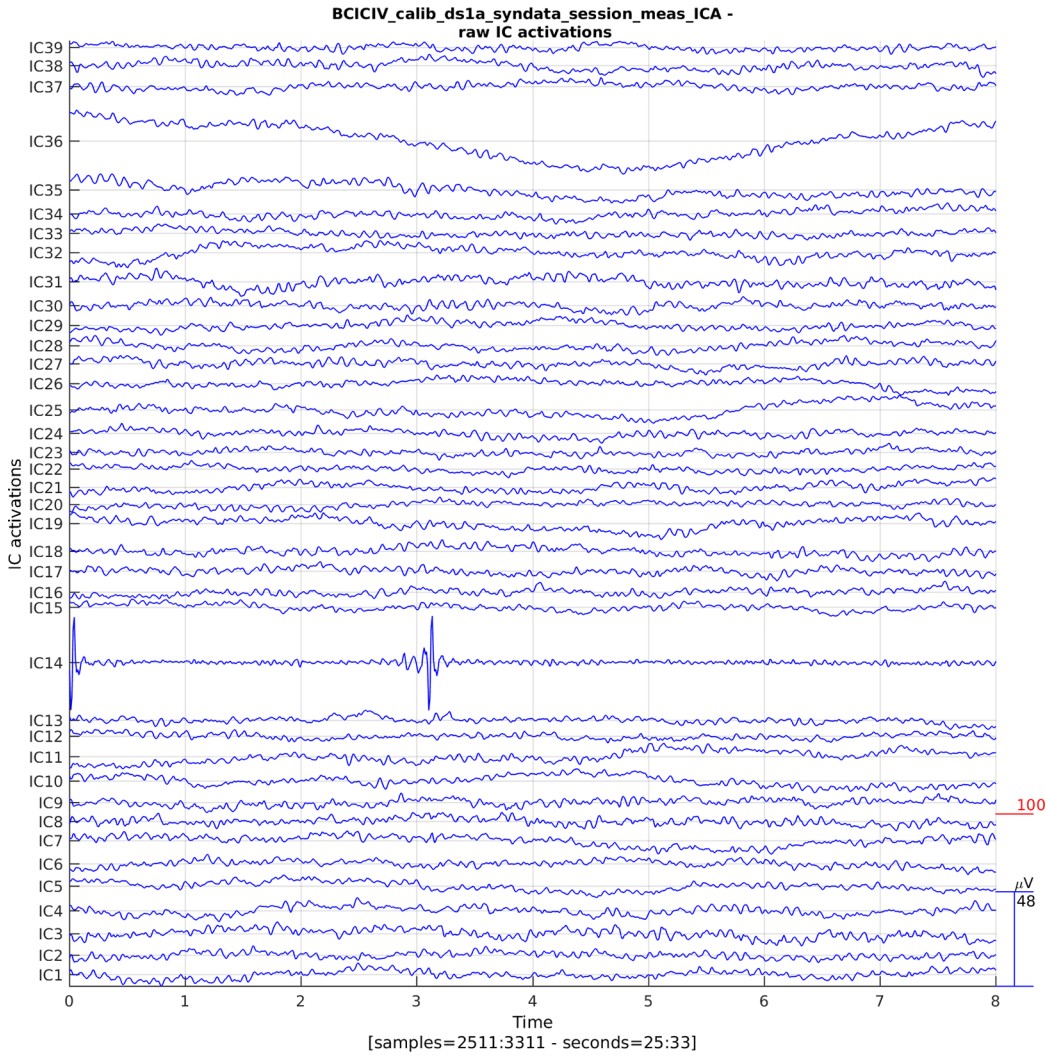

**Figure 11 Plot of raw IC activations after all processing steps.** IC14 shows clear evidence of EMG noise remaining; while IC36 may indicate a drifting channel.

required set of functions directly, has several benefits. Function names and parameters become objects available for later processing, so one can operate on them, e.g. to record what was done to logs and to swap functions/parameters on the fly or to check the specification of the pipe. By specifying new approaches in new pipe files, and saving already-tried pipe files, one can treat the *files* as a record of attempted preprocesses. This record corresponds to the user's perspective, and thus complements the additional history structure saved to the EEG file, which records all parameters for each operation not only those specified by the user. Finally, the user should not usually rely on defaults (as given by CTAP, EEGLAB or other toolboxes), because the optimal choice often depends on the data. This is also one reason to separately define pipeline and parameters. Separating these as objects is convenient for e.g. testing multiple parameter configurations. A single script file per analysis approach is incompatible with parameter

optimisation, because the number of different possible combinations begins to require a layer of code to manage the scripts—this is exactly what CTAP provides.

As different analysis strategies and methods can vary greatly, CTAP was implemented as a modular system. Each analysis can be constructed from discrete steps which can be implemented as stand-alone functions. As CTAP is meant to be extended with custom analysis functions the interface between core CTAP features and external scripts is well defined in the documentation. The only requirement is to suppress any pop-ups or GUI-elements, which would prevent the automatic execution of the analysis pipe.[11] It is also up to the user to call the functions in the right order.

The system supports branching. This means that the analysis can from a tree-like structure, where some stage is used as input for multiple subsequent workflows. To allow this, any pipe can act as a starting point for another pipe. The CTAP repository provides a simple example get the user going. For branches to appear, a bare minimum is a collection of three pipes of which one is run first. The other two both act on this output but in different ways. Currently the user is responsible for calling the pipes of a branched setting in a meaningful order. However, this is straightforward to implement and having the analysis logic exposed in the main batch file makes it, e.g. easy to run only a subset of the branches.

Although CTAP works as a batch-processing pipeline, it supports seamless integration of manual operations. This works such that the user can define a pipeline of operations, insert save points at appropriate steps and work manually on that data before passing it back to the pipe. The main extra benefit that CTAP brings is to handle bookkeeping for all pipeline operations, such that manual operations become exceptional events that can be easily tracked, rather than one more in a large number of operations to manage.

Computational testing for automated preprocessing never overrides the user's configuration options, even when these might break the pipe. For example, `CTAP_reject_data.m` contains code to auto-detect the data to reject. However, the user can set this option explicitly, and can do so without having first called any corresponding detection function, which will cause preprocessing on that file to fail. Allowing this failure to happen is the most straightforward approach, and ultimately more robust. Combined with an informative error message the user gets immediate feedback on what is wrong with the pipe.

On the other hand, CTAP does provide several features to handle failure gracefully. As noted, the pipe will not crash if a single file has an unrecoverable error, although that file will not be processed further. This allows a batch to run unsupervised. Then, because no existing outputs are overwritten automatically, one can easily mop-up the files that failed without redoing all those that succeeded, if the fault is identified. Because pipes can be divided into step sets, tricky processes that are prone to failure can be isolated to reduce the overall time spent on crash recovery. CTAP saves crashed files at the point of failure (by setting the parameter 'trackfail' in `CTAP_pipeline_looper.m`), permitting closer analysis of the problematic data.

In contrast to many analysis plug-ins built on top of EEGLAB, no GUI was included in CTAP. While GUIs have their advantages (more intuitive data exploration, easier

---

[11] As noted above, for this reason much original code has been refactored to avoid runtime-visible or focus-grabbing outputs. The ultimate aim is for CTAP to interface directly to Matlab functions to remove dependency on EEGLAB releases, while retaining compatibility with the EEGLAB data structure.

for novice users, etc.) there is a very poor return on investment for adding one to a complex batch-processing system like CTAP. A GUI also sets limits to configurability and can constrain automation if CTAP is executed on a hardware without graphical capabilities. The absence of GUI also makes the development of extensions easier as there are fewer dependencies to handle.

In contrast to many other broad-focus physiological data analysis tools, CTAP is designed to meet a very focused goal with a specific approach. This does however create some drawbacks. Compared to scripting one's own pipeline from scratch, there are usage constraints imposed by the heavy use of struct-passing interfaces. Some non-obvious features may take time to master, and it can be difficult (albeit unnecessary) to understand the more complex underlying processes.

Computational testing for automated preprocessing is also built to enable easy further development by third parties, by using standardised interfaces and structures. This was a feature of original EEGLAB code, but contrasts with many of the EEGLAB-compatible tools released since, whose functionality was often built-in an *ad hoc* manner. The main requirement for development is to understand the content and purpose of the `EEG.CTAP` field (which is extensively documented in the wiki), and the general logic of CTAP.

Developers can easily extend the toolbox by using (or emulating) the existing `ctapeg\_*.m` functions, especially the `ctapeg_detect_*.m` functions, which are simply interfaces to external tools for detecting artefacts. Existing `CTAP_*.m` functions can be relatively more complex to understand, but the existing template provides a guideline for development with the correct interface.

## Future work

Computational testing for automated preprocessing is far from finalised, and development will continue after the initial release of the software. The main aim of future work is to evolve CTAP from workflow management towards better automation, with computational comparative testing of analysis methods, to discover optimal parameters and help evaluate competing approaches.

As stated above, the potential to fully automate EEG processing is constrained by the indeterminacy of EEG: known as the *inverse problem*, this means that it is not possible to precisely determine a ground-truth for the signal, i.e. a unique relationship to neural sources. The signal can also be highly variable between individuals, and even between intra-individual recording sessions (*Dandekar et al., 2007*). These factors imply that there cannot be a general algorithmic solution to extract neurally generated electrical field information from EEG, thus always requiring some human intervention. By contrast, for example in MEG certain physical properties of the system permit inference of sources even from very noisy data (*Taulu & Hari, 2009*) (although recording of clean data is always preferable, it is not always possible, e.g. with deep brain stimulation patients (*Airaksinen et al., 2011*)).

While many publications have described methods for processing EEG for different purposes, such as removing artefacts, estimating signal sources, analysing ERPs and so on. However, despite the wealth of methodological work done, there is a lack of

benchmarking or tools for comparison of such methods. The outcome is that the most reliable way to assess each method is to learn how it works, apply it and test the outcome on one's own data: this is a highly time-consuming process which is hardly competitive with simply performing the bulk of preprocessing in a manual way, as seems to remain the gold standard. The effect of each method on the data is also not commonly characterised, such that methods to correct artefacts can often introduce noise to the data, especially where there was no artefact (false positives).

Thus, we also aim to enable testing and comparison of automated methods for preprocessing. This is still work in progress, as we are building an extension for CTAP that improves testing and comparison of preprocessing methods by repeated analyses on synthetic data. This extension, tentatively titled Handler for sYnthetic Data and Repeated Analyses (HYDRA), will use synthetic data to generate ground-truth controlled tests of preprocessing methods. It will have capability to generate new synthetic data matching the parameters of the lab's own data, and compare outcomes of methods applied to this data in a principled computational manner. This will allow experimenters to find good methods for their data, or developers to flexibly test and benchmark their novel methods.

Another desirable, though non-vital, future task is to expand the quality control output, to include functionality such as statistical testing of detected bad data, for the experimenter to make a more informed decision. Although statistical testing is already implied in many methods of bad data detection, it is not visible to users. This will take the form of automated tools to compare output from two (or more) peeks, to help visualise changes in both baseline level and local wave forms.

Such aims naturally complement the work of others in the field, and it is hoped that opportunities arise to pool resources and develop better solutions by collaboration.

## CONCLUSION

The ultimate goal of CTAP is to improve on typical ways of preprocessing high-dimensional EEG data through a structured framework for automation.

We will meet this goal via the following three steps: (a) facilitate processing of large quantities of EEG data; (b) improve reliability and objectivity of such processing; (c) support development of smart algorithms to tune the thresholds of statistical selection methods (for bad channels, epochs, segments or components) to provide results which are robust enough to minimise manual intervention.

We have now addressed aim (a), partly also (b) and laid the groundwork to continue developing solutions for (c). Thus, the work described here provides the solid foundation needed to complete CTAP, and thereby help to minimise human effort, subjectivity and error in EEG analysis; and facilitate easy, reliable batch-processing for experts and novices alike.

## ACKNOWLEDGEMENTS

The authors would like to thank Andreas Henelius, Miika Toivanen, Kristian Lukander and Lauri Ahonen for fruitful discussions on the CTAP toolbox and this paper.

### Funding

This work was partly supported by the Revolution of Knowledge Work project no. 40228/13, funded by Tekes—the Finnish Funding Agency for Technology and Innovation. The funders had no role in study design, data collection and analysis, decision to publish or preparation of the manuscript.

### Grant Disclosures

The following grant information was disclosed by the authors:
Revolution of Knowledge Work project no. 40228/13, funded by Tekes—the Finnish Funding Agency for Technology and Innovation.

### Competing Interests

The authors declare that they have no competing interests.

### Author Contributions

- Benjamin U. Cowley conceived and designed the experiments, performed the experiments, analysed the data, wrote the paper, prepared figures and/or tables, performed the computation work and reviewed drafts of the paper.
- Jussi Korpela conceived and designed the experiments, performed the experiments, analysed the data, wrote the paper, prepared figures and/or tables, performed the computation work and reviewed drafts of the paper.
- Jari Torniainen conceived and designed the experiments, performed the experiments, analysed the data, prepared figures and/or tables, performed the computation work and reviewed drafts of the paper.

### Data Deposition

https://github.com/bwrc/ctap.

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
