# Peer review of "Computational testing for automated preprocessing: a Matlab toolbox to enable large scale electroencephalography data processing"

_PeerJ Computer Science, doi:10.7717/peerj-cs.108_

## Round 0.1 · original submission · Major Revisions

Two of the three reviewers recommend a major revision, while one reviewer recommends rejection.

Reviewer #1 structured the review in a manner that interleaves some of the review guidelines with the actual reviewer's comments. (This should be clear from the context.) Reviewer #2, while recommending to reject the article, does provide many useful and helpful comments for improvement. Therefore please also consider this review carefully and respond to the individual comments and suggestions. Similarly, Reviewer #3 provides numerous detailed suggestions for improvements.

After considering all reviews carefully, on the balance of it, I think the paper can be salvaged and a major revision accepted, provided the key concerns identified by the reviewers are addressed appropriately. These include some corrections and minor clarifications (e.g., several reviewers point out that EEGLAB can be used without a GUI from the CLI or script; see reviews for details). However, the bulk of the additional effort would have to be spent towards the major concerns raised by the reviewers. These include:

questions raised by Reviewer #2 on the IIR and FIR filters, and related items (under "Validity of Findings"); and
re-focusing the description and evaluation on what is novel or unique about the proposed solution.

Please consider all three reviews carefully when preparing your revision and provide responses to the comments and suggestions by the reviewers, clearly indicating how you have addressed the various concerns.

Reviewer 1 ·

Basic reporting

The submission must adhere to all PeerJ Computer Science policies.
The submission meet to all PeerJ Computer Science policies.

The article must be written in English using clear and unambiguous text and must conform to professional standards of courtesy and expression.
The article should be revised for style, grammar and typos. The authors should avoid the use of technical jargons

The article should include sufficient introduction and background to demonstrate how the work fits into the broader field of knowledge. Relevant prior literature should be appropriately referenced.
The authors use abundant literature in the manuscript. However, they should provide more relevant references and insert a section for already existing automated MATLAB-Octave pipelines (See. http://psom.simexp-lab.org/).

The structure of the submitted article should conform to one of the templates. Significant departures in structure should be made only if they significantly improve clarity or conform to a discipline-specific custom.
No comments

Figures should be relevant to the content of the article, of sufficient resolution, and appropriately described and labeled.
Labeling of figures and explanations needs important improvements. The authors should provide labels and descriptions that make the figure a self-contained unit.
e.g.
Figure 2: Definitions and abbreviations are not explained
Figure 3: The authors does not describe or provide relevant information on the figure
Figure 4: Size of figure should be reduced and typo on word ‘horizontal’ must be fixed. X label is missing in the figure.
Figure 6: The authors should include a legend in the figure and define or remove abbreviations as ‘pdf’ or ‘sd’

The submission should be ‘self-contained,’ should represent an appropriate ‘unit of publication’, and should include all results relevant to the hypothesis. Coherent bodies of work should not be inappropriately subdivided merely to increase publication count.
In the manuscript the authors reference the wiki page developed for the toolbox to avoid deeper explanations on the matter of discussion. This should be avoided as this affect the unity and ‘self-containing’ aspect of the manuscript.

All appropriate raw data has been made available in accordance with our Data Sharing policy.
No comments

Formal results should include clear definitions of all terms and theorems, and detailed proofs.
No comments

Experimental design

The manuscript under revision propose an automated pipeline for preprocessing and compare different streamings of analysis for electroencephalographic (EEG) data. The development of the toolbox , Computational Testing for Automated Preprocessing (CTAP), is proposed as an extension of the already existing and popular EEGLAB Toolbox (Delorme and Makeig, 2004) developed at the Swartz Center for Computational Neurosciences at UCSD. While it is nice to see a new contribution to the EEGLAB community, it is necessary to make some important points on the proposed manuscript/development.
In the manuscript, when reviewing the previous work on the field, the authors suggest that EEGLAB is a GUI based tool with restricted ability to be used for batch or custom data analysis scripts. Later the authors use this to justify the development of the CTAP. However, the reality is that EEGLAB provide both GUI and scripting based capabilities. The authors are aware of this, since these scripting capabilities made possible their own development using EEGLAB as a base, as well as the many contribution received as EEGLAB's extension or plugins. This should be mentioned in the manuscript.

Regarding the relevance of the contribution. At the beginning of the manuscript, the authors claim to propose a pipeline to compare different streamings of EEG data analysis. This is a very good idea and and also a need of the EEG community. However, what is presented instead is a framework that comes on top of EEGLAB scripting capabilities (core and plug-in), leaving the main and most exciting part of the contribution for a future work.

I understand that the development of this framework is necessary for the later development of a functionality to compare different streaming of preprocessing, however, without this ‘comparing’ functionality, the contribution of the current work is reduced to a new wrap on an already existing functionality in EEGLAB.

Validity of the findings

No Comments

Additional comments

As mentioned before, the idea of providing and environment for comparison of different processing streams of EEG data is bold and would be an important contribution to the EEG community. However, this contribution is not included in this manuscript. Apparently, a few scripts could do the trick. Given the ability to do exactly the same thing as the CTAP does with just a bit of scripting using EEGLAB, the authors should consider to hold on the submission and make it more valuable by implementing the ideas enunciated in the section ’Future work’.

The authors should also consider an expanded view on artifact rejection and asses not only for eye blinks but for all the many artifacts the EEG data is sensitive to.

I sincerely encourage the authors to continue working on this development and make a new submission of the manuscript attending to the formulated criticisms.

Reviewer 2 ·

Basic reporting

The manuscript „Computational Testing for Automated Preprocessing: a Matlab toolbox for better electroencephalography data processing“ by Cowley and colleagues describes the general structure and workflow of a MATLAB toolbox for EEG data processing. The toolbox extends the widespread EEGLAB MATLAB toolbox by an additional command line oriented user interface optimized for batch pre-processing of EEG data and a generator EEG test signals.

Language and writing are accessible and appropriate. The manuscript does not present original research but the description and documentation of a software (interface) for more efficient data analysis. I’m not sure whether the manuscript is within the scope of the journal. In general, most of the content of the manuscript would be expected in a software manual rather than in a research article. The manuscript could be massively improved if the focus is moved away from the mere documentation of the software (interface and workflow) towards the genuine scientific contribution, e.g. invention and/or implementation of new algorithms of signal processing and classification. Furthermore, the manuscript could be improved by adding empirical data. E.g., trying to demonstrate empirically that the approach and the implemented algorithms (e.g. blink detection and classification etc.) indeed enhance the quality of the pre-processed data.

From the information provided in the manuscript I cannot follow the claim in the abstract that the toolbox helps avoiding manual decision making to reduce subjectivity and low replicability. I do not see how the software reduces manual decision making except providing default values for some pre-processing procedures. The general validity of default values in such a diverse field as psychophysiology is questionable. Furthermore, the abstract promises testing and comparison methods. This appears to refer mainly to synthetic test signals but not to real world data. While I see the requirement to evaluate pre-processing procedures with test signals and do see the benefits of standardized and replicable test signals-as provided by the toolbox-I do not see how the toolbox helps or enforces validation with real data.

Other:
Line 22: “by” missing?
Line 62: The chapter does not provide anything essential for the manuscript. Considerably shorten? Omit?
Line 77-78: This statement is incorrect. The GUI is part of EEGLAB but the software can be used completely without from CLI or script (as the authors actually do and thus, should know).
Line 89: BrainVISION Analyzer
Figures 4 and 6: Units missing or incorrect?!

Experimental design

No experimental work is described in the manuscript.

Validity of the findings

No new findings are described in the manuscript.

An in-depth review of the toolbox code is not possible within the scope of a manuscript review. For curiosity I had a look at minor parts of the code, more or less accidentally starting with data filtering. The default IIR elliptic filter is not suitable as a general purpose filter for EEG data. The IIR filter is not part of the default EEGLAB toolbox but a plugin which appears to be no longer maintained since quite some time and known to be problematic in several aspects. The alternative EEGLAB FIR filter routines used by the toolbox are known to be broken at least since 2012 (Widmann et al., 2012, DOI: 10.3389/fpsyg.2012.00233), should no longer be used and were replaced. The documented equation for default filter order is incorrect (in particular for the IIR but also for the FIR filter). The defaults for the cutoff frequencies are inappropriate for general purpose EEG analysis (see e.g. Acunzo et al., 2012, DOI: 10.1016/j.jneumeth.2012.06.011; Tanner et al., 2015, DOI: 10.1111/psyp.12437).

Additional comments

I sympathetically see the requirement to supplement such a toolbox with a peer-review publication. However, I would suggest to replace the rather documentational aspects (much better fitting into a software manual) by (a) empirical validation of the toolbox and (b) focus on the description and evaluation of the genuine scientific contribution (e.g., blink detection, detailed description and evaluation of the validity of the synthetic test signals, description and evaluation of provided defaults).

·

Basic reporting

Overall, the paper would be clearer if edited in plain English as much as possible, and using examples to clarify many unclear and often unnecessary technical terms. I get what you’re trying to do, and this is important work, but the paper reads as a rough draft at the moment: it really needs to be restructured and edited.

There are typos throughout - please edit carefully.

Avoid the quotation marks of mystery and be specific instead, for instance in:
complex ’standard’ EEG processing
’quality control’
of ’bad’ data
’mop up’ 
’tricky’
’gold standard’

Avoid monstrosities such as “learning curve”.

## abstract
“the pre-processing of EEG data is quite complicated”. Unless you provide an example, it remains unclear why this is the case. It would be fair to say that there are many options, but you can see certain large effects without pre-processing at all - which is certainly not the case for other brain imaging techniques.

## Introduction
Overall, the introduction would be much better restructured by pointing out the main steps involved in EEG preprocessing, instead of hinting at its complexity, as well as existing tools in Matlab & Python. Then describe what’s lacking and your contribution. At present, you ignore all the pipeline toolboxes, and you fail to precisely describe what is lacking.

This sentence is difficult to parse and refers to undefined “electrophysiology data”: “However, among those types of human electrophysiology data recorded from surface electrodes (to which EEG is most similar in terms of recording methods, see e.g. Cowley et al. (2016) for a review), EEG data is comparatively difficult to pre-process.”
EEG is difficult to pre-process compare to what?

The class A and class B terminology does not bring anything: use clear sub-headings instead, and focus on examples, to avoid technical sounding but vague terms.

“EEG data is high-bandwidth” -> EEG data can be very large.

“Due to the inverse problem it is not possible to
precisely determine a ’ground truth’ for the signal”
It’s unclear how this relates to problems with preprocessing.

## Related work
This section comes far too late and should be integrated in a more comprehensive introduction.

“EEGLAB is a graphical user interface (GUI)-based tool” - this is inaccurate, because EEGLAB has been developed to be used at either GUI or script level.

I don’t see the point of Figure 1: it is rather confusing. The terms would be better defined in the text.

Experimental design

“CTAP is built on Matlab (r2015a and higher)” - does it mean it was tested with 2015a and higher, or absolutely not usable with 2014b and under?
Same question with EEGLAB.

## Approach
“Although CTAP works as a batch processing pipeline, it supports seamless integration of manual operations. This works such that the user can define a pipeline of operations, insert save points at appropriate steps, and work manually on that data before passing it back to the pipe.”
This can be done easily using EEGLAB + scripts, and is covered in the EEGLAB tutorial. How does your approach differ? Also, how does it differ from PREP and other tools?

## Methods
After explaining that CTAP differs from existing toolboxes, the methods section starts with:
“The core activity of CTAP is preprocessing EEG data by cleaning artefacts, i.e. detection and either correction or removal of ’bad’ data, that is not likely to be attributable to neural sources.”
which is what existing toolboxes do. So again, the introduction must do a much better job at clarifying what is new here.

## Configuration
In the 2 step example, please do not use *i* as an index, as in:
`i = 1; %stepSet 1`
`s` or `step` would be better coding.

## Pipe execution
Can you provide an estimate of the space needed to store the typical directory tree, given that all intermediate stages are saved? I realise this will depend on epoch lengths and if time-frequency decomposition is performed.

## CTAP outcomes
For some of the main outcomes, such as blinks, bad segments, ICA, please clarify which tools are used, for instance from EEGLAB exclusively? I see that the functions and algorithms are described later, in results. These explanations should be part of methods.

Validity of the findings

## Results
Overall, the result section would be more convincing if it provided a more detailed walkthrough of the preprocessing of a dataset, focusing on the figure outputs.

One important aspect is missing, which is promised in the abstract: “testing and comparison of automated methods”. Can you explicitly describe how this can be done using CTAP? Do you have special functions for instance to generate figures comparing blink correction techniques? The discussion mentions that such features are not yet available, so part of the software description, including in the abstract, is misleading.

Please define “trimmed sd”. Do you mean the SD of the trimmed mean. If so, what amount of trimming is applied?

## Discussion
I agree with your points about GUIs and software development.

---

## Round 0.2 · accepted · Accept

Thank you for your detailed rebuttal letter and addressing the original reviewers' comments.

·

Basic reporting

I'm satisfied with the changes.

Experimental design

I'm satisfied with the changes.

Validity of the findings

I'm satisfied with the changes.

Additional comments

Your new version is much better - good job integrating all the comments.
I hope your toolbox will contribute to more transparent and reproducible EEG analyses.
Happy software development in 2017!